# Aristolochic Acid I-Induced Hepatotoxicity in Tianfu Broilers Is Associated with Oxidative-Stress-Mediated Apoptosis and Mitochondrial Damage

**DOI:** 10.3390/ani11123437

**Published:** 2021-12-02

**Authors:** Dan Xu, Lizi Yin, Juchun Lin, Hualin Fu, Xi Peng, Lijen Chang, Yilei Zheng, Xiaoling Zhao, Gang Shu

**Affiliations:** 1Farm Animal Genetic Resources Exploration and Innovation Key Laboratory of Sichuan Province, Sichuan Agricultural University, Chengdu 611130, China; 13086605395@163.com (D.X.); zhaoxiaoling@sicau.edu.cn (X.Z.); 2Department of Basic Veterinary Medicine, Sichuan Agricultural University, Chengdu 611130, China; yinlizi@hotmail.com (L.Y.); juchunlin@126.com (J.L.); fuhl.sicau@163.com (H.F.); 3Sichuan Industrial Institute of Antibiotics, Chengdu University, Chengdu 611130, China; pengxi197313@163.com; 4Department of Veterinary Clinical Science, College of Veterinary Medicine, Oklahoma State University, Stillwater, OK 74078, USA; lj.chang@okstate.edu; 5College of Veterinary Medicine, University of Minnesota, Minneapolis, MN 55791, USA; zhen0219@umn.edu

**Keywords:** aristolochic acid, liver, oxidative stress, apoptosis, mitochondria

## Abstract

**Simple Summary:**

Aristolochic acid (AA) is a component of traditional Chinese herbs and commonly used in the farm poultry industry in China for anti-infection, anti-viral and anti-bacterial treatment. However, long-term and over-exposure of these drugs has been proven to be associated with serious hepatotoxicity, but the mechanism of AA-I-induced hepatotoxicity remains unknown. Therefore, in this study, a subchronic toxicity test was conducted to evaluate the mechanism of AA-I-induced hepatotoxicity in Tianfu broilers. Subchronic exposure to high doses of AA-I in broilers can cause serious hepatotoxicity by breaking the redox balance to form oxidative stress, along with promoting oxidative-stress-mediated apoptosis and mitochondrial damage. In conclusion, AA-I has been found to damage broilers’ livers in high doses. This study provides suggestions for the clinical application of traditional Chinese medicine containing AA-I in the poultry industry.

**Abstract:**

Aristolochic acid (AA) is a component of traditional Chinese herbs and commonly used for farm animals in China. Over-exposure of AA has been proven to be associated with hepatotoxicity; however, the mechanism of action of AA-I-induced hepatotoxicity remains unknown. In the current study, a subchronic toxicity test was conducted to evaluate the mechanism of AA-induced hepatotoxicity in Tianfu broilers. According to the results, AA-I-induced hepatotoxicity in Tianfu broilers was evidenced by the elevation of liver weight, levels of serum glutamic oxalacetic transaminase (GOT) and glutamic-pyruvic transaminase (GPT). Furthermore, hepatocyte swelling, vesicular degeneration and steatosis were observed. Additionally, AA-I elevated the production of reactive oxygen species (ROS) and induced oxidative stress, which further led to excessive apoptosis, characterized by mitochondrial depolarization, upregulation of Bax, and down-regulation of Bcl-2 expression. In conclusion, the mechanism of AA-I-induced hepatotoxicity was associated with oxidative-stress-mediated apoptosis and mitochondrial damage.

## 1. Introduction

Aristolochic acids (AAs) are chemical components of various natural herbs, especially in Aristolochiaceae [1]. AA exerts various therapeutic effects, including anti-neoplasia, anti-infection, anti-inflammatory, analgesia and anti-fertility [2]. Therefore, AA has been used extensively for more than 2500 years in many countries as a herbal medication to treat different disease, such as eczema, pneumonia, stroke, hepatitis, snake bites, arthritis and gout [3,4]. AA-I (8-methoxy-3,4-methylenedioxy-10-nitrophenanthrene-1-carboxylic acid, Figure 1) has been recognized as the main toxic component of AA [5], inducing aristolochic acid nephropathy (AAN) [6]. Researchers have indicated that AAN is manifested by rapidly progressive interstitial nephropathy [7], and is thought to occur through multiple mechanisms, such as renal cell apoptosis, renal interstitial inflammatory cell infiltration, oxidative stress and DNA adducts [8,9]. Recently, AA-I-induced DNA adduct accumulation was detected in the liver, which implies that AA-I can induce hepatotoxicity [10,11]. At the same time, many reports have evidenced the direct association between the use of AA and hepatitis-B-related liver cancer [12]. Although the renal toxicity of AA-I has been clearly documented to date, information regarding AA-I-induced hepatotoxicity remains scant and unclear. The liver plays an important role in drug and toxin metabolism [13]; thus, the mechanism of action of AA-I-induced liver injury should be identified.

The use of botanical drugs containing AA has been prohibited in many countries due to its toxic effects [14]. However, there are three drugs derived from the Aristolochia family, including Aristolochia, Herba Aristolochiae, and Asarum heterotropoides that remain in the Pharmacopoeia of PR China for clinical use to date as complementary remedies for poultry husbandry, especially used as a feed additive in domestic poultry, leading to a potential hazard to domestic poultry husbandry [15]. Moreover, a recent study also indicated that AA-I could move from Aristolochia family herbs to other plant species via a common matrix: the soil, which may make it easier for animals to be exposed to AA-I [16]. Therefore, it is necessary to recognize the toxicity of AA-I systemically, thus providing appropriate guidance regarding the clinical use of AA-I.

Exogenous chemicals can be metabolized to active metabolites by hepatic enzymes in the liver, which may further induce free radicals and lead to peroxidation. Hepatocyte apoptosis and necrosis can be triggered by the aforementioned active metabolites, leading to hepatic injury and hepatic carcinogenesis [17]. Moreover, most of the complications associated with hepatic injury are mediated by oxidative stress, oxidative-stress-related mediators and inflammation [18]. It has been indicated that oxidative stress, apoptosis and inflammation play important roles in AA-induced toxicity [19,20,21].

The LD50 of AA-I to induce subchronic toxicity in Tianfu broilers has been determined in a previous study [15]. The aim of this study is to explore the mechanism of AA-I-induced hepatotoxicity, including oxidative stress, oxidative-stress-mediated apoptosis and mitochondrial damage, in order to provide guidance for the application of AA-I in domestic poultry husbandry.

## 2. Material and Methods

### 2.1. Experimental Materials

Aristolochic acid A (AA, HPLC > 98%) was purchased from Chengdu Rifens Tech-nology Co., Ltd., Sichuan, China. TRIzol reagent (Nanjing Jiancheng Bioengineering Institute, Nanjing, China), total ROS assay kit (Thermo Fisher, Waltham, MA, USA), MMP detection JC-1 kit (BD Bioscience, Franklin Lakes, NJ, USA), eBioscienceTM Annexin V-FITC Apoptosis Kit (Invitrogen, Shanghai, China), and reverse transcription reagents (TaKaRa, Kyoto, Japan) were purchased from the relevant companies. The chicken-specific ELISA assay kits for glutamic oxalacetic transaminase (GOT), glutamic-pyruvic transaminase (GPT), total antioxidant capacity (T-AOC), malondialdehyde (MDA), superoxidase dismutase (SOD) and glutathione (GSH) were purchased from Nanjing Jiancheng Bioengineering Institute (Nanjing, China).

### 2.2. Experimental Animals

The study has been approved by the Institutional Animal Care and Use Committee of Sichuan Agricultural University (permission number DYY-2018203007) and was conducted at the poultry farm of Sichuan Agricultural University. Animals and procedures were conducted in accordance with the regulations of the national standard Laboratory Animal Requirements of Environment and Housing Facilities (GB 14925–2001).

Forty, 1-day-old, male Tianfu broilers with body weight of 38.83 ± 3.45 g from the Poultry Breeding Farm of Sichuan Agricultural University (Sichuan, China) were used in this study. All subjects were captive in a climate-controlled facility with air-ventilator and the relative humidity was kept at 50%. The photoperiod was maintained at 24 h for the first 14 days, followed by 20 h for the rest of the study. The temperature was set at 34 °C initially, followed by decremented 2 °C per week until the temperature reached 26 °C, which was maintained throughout the study. Food and water were free to access throughout the study. The daily diet formulation was determined according to the National Research Council requirements for chickens (Appendix A) [22].

### 2.3. Subchronic Poisoning with Aristolochic Acid A in Tianfu Broilers

Three different doses (1/100, 1/50, and 1/10) of LD50, with these amounts determined in a previous study [15], were used to induce 28-day subchronic toxicity tests in broilers to evaluate the mechanism of AA-I-induced biochemical profile and liver histopathological changes.

Forty, 1-day-old, male broilers were randomly divided into four groups. Subjects in the control group (CG) received normal saline. AA-I at 1/100, 1/50, and 1/10 of LD50 was administered to subjects in the low-dose group (LAG), middle-dose group (MAG), and high-dose group (HAG), respectively. Drugs were administered intraperitoneally in all groups. Subjects were acclimated in the facility for 24 h (D 0), followed by the administration of experimental agents once daily (q 24 h) for 28 days (D 1 to D 28).

#### 2.3.1. Sample Collection and Liver Cell Separation

All subjects were sacrificed, and samples were collected on D 29. The liver was weighed and the ratio of liver weight to body weight (g/kg) (relative liver index) was calculated.

Liver samples were fixed in 4% (*w*/*v*) buffered paraformaldehyde for histology. The remaining liver samples were flushed by 0.9% normal saline for biochemical analysis. The harvested hepatic cells were used for flow cytometry analysis. The method of harvesting hepatic cells has been described elsewhere [17].

#### 2.3.2. Biochemical Analysis

The levels of serum GOT, serum GPT, and hepatic MDA, GSH, T-AOC and SOD were detected by specific ELISA kits. The method of biochemical analysis has been described in a previous study [23].

#### 2.3.3. Histopathology

The method for detection of histological changes has been described in a previous study [24]. The fixed samples were dehydrated, cleaned, embedded in paraffin, and sliced to 5 μm slices by use of an RM2235 microtome (Leica, Munich, Germany). Sliced samples were flattened and dried on glass slides, stained with hematoxylin and eosin (Thermo, Waltham, MA, USA), and sealed with neutral resin. The histopathological changes were observed by a CX22 microscope (Olympus, Tokyo Metropolitan, Japan) and a DM1000 microimaging system (Leica, Munich, Germany) was applied for image recording. In pathological diagnosis, we used INHAND criteria (Table 1) to judge the severity of histological lesions [25].

#### 2.3.4. Reactive Oxygen Species (ROS), Mitochondrial Membrane Potential (MMP) and Apoptosis by Flow Cytometry

The prepared hepatocytes were incubated with DCFH-DA (total ROS assay kit), JC-1, and 5 μL of Annexin V-FITC and 5 μL of PI, respectively. All hepatocytes were incubated in a dark environment at 37 °C for 15 min. A cytometry flowmeter (Bio-Rad, Hercules, CA, USA) was used to detect reactive oxygen species (ROS, %), mitochondrial membrane potentials (MMPs) and the percentage of apoptosis. The MMP result was described as a mitochondrial depolarization ratio. All aforementioned data were translated and recorded by Kaluza 2.1 Software, Beckman Coulter, CA, USA.

#### 2.3.5. Ultrastructure Observations

The observation criteria of ultrastructure were published elsewhere [26]. Liver samples were sliced and fixed in 2.5% glutaraldehyde (pH = 7.4) at 4 °C, followed by osmium tetroxide (OsO4) for post-fixation. The samples were soaked and embedded in epoxy resin acetone solution once dried. The samples were sliced to 80 nm slices by Microtome. Samples were stained by lead citrate and uranyl acetate solutions. Ultrastructure of ileum was observed by an HT7700 transmission electron microscope (Hitachi, Tokyo, Japan) and photographs were taken by a GANTAN830.10W CCD camera (Hitachi, Tokyo, Japan). In pathological diagnosis, we used INHAND criteria (Table 1) to judge the severity of histological lesions [25].

#### 2.3.6. Fluorescence Real-Time Quantification PCR (QRT-PCR)

Samples were rinsed by pre-frozen Diethypyrocarbonate (DEPC) water and then frozen in liquid nitrogen at −80 °C. Approximately 60 mg of frozen sample was ground thoroughly in a precooling tissue homogenizer. Total RNA was extracted by a commercial kit (TRIzol. Invitrogen, Shanghai, China). The RNA concentration was detected by a nucleic acid protein analyzer which had the D260/D280 reverse transcription eligible range at 1.8–2.0. The cDNA was stored at −80 °C.

According to the operator’s manual, QRT-PCR could be used to detect the expression of Nrf2, NQO1, HO-1, Bax and Bcl-2, and β-actin could be used for normalizing the expression of aforementioned genes as an endogenous control. In this study, the expression of the aforementioned genes in CG was defined as baseline (value = 1), and the 2−ΔΔCt method was applied for quantifying gene expressions, where ΔΔCt is the detected ΔCt- reference ΔCt and ΔCt is the Ct _target gene_ − Ct _housekeeping gene_. Primers used in the present study (Appendix A) were designed by Premier 5 (PREMIER Biosoft International, New York, NY, USA) and synthesized by Chengke BioTech Co., Ltd. (Guangzhou, China).

### 2.4. Statistical Analyses

All parameters were presented as mean ± standard error (mean ± SE) in this study. Duncan’s test was used to determine the differences among treatment groups. Data were analyzed by one-way analysis of variance (ANOVA) with SPSS 19.0 (SPSS Inc., Chicago, IL, USA). Significance was determined when *p* < 0.05.

## 3. Results

### 3.1. AA-I Elevated Liver Index in Broilers

The macroscopic characteristics of liver samples are shown in Figure 2A. Samples in CG were normal under gross inspection with red-brown, bright and damp surfaces, whereas samples in treatment groups, especially in MAG and HAG, showed swollen, firm, fragile grossly and diffused yellowish-brown steatosis lesions. Compared with the CG, the liver index in MAG and HAG increased significantly (*p* < 0.05) after intraperitoneal administration of AA-I for 28 days. The liver index results are shown in Figure 2B.

### 3.2. AA-I Elevated Hepatic Enzymes in Broilers

The serum transaminase levels are shown in Figure 3. Compared with the CG, all AA-I treatment groups recorded significant higher levels of serum GOT (*p* < 0.05), and nearly doubled levels were observed in MAG and HAG (*p* < 0.05). For serum GPT, both MAG and HAG showed significant elevations of serum GPT in comparison with that of CG (*p* < 0.05).

### 3.3. AA-I Deformed Hepatocyte Structures

The results of the histopathological changes of hepatocytes are shown in Figure 4. Hepatocytes in CG were normal in structure and cellular arrangements. AA-I-induced hepatocyte degeneration and necrosis in all intervention groups presented in a dose-dependent manner. Adipose changes and the cloudy swelling of hepatocytes were recognized by observations of red granular substances, irregular or circular vacuoles in the cytoplasm. Necrotic hepatocytes could be identified by pyknosis, karyorrhexis, and karyolysis. In LAG, mild hepatocyte degeneration and mild narrowing of hepatic sinuses were observed. In the MAG, moderate histopathological lesions of the liver were recognized by adipose degeneration, hydropic degeneration and the disappearance of hepatic sinuses. The severest lesions were found in the HAG, where hepatocytes showed severe swelling, vesicular degeneration and steatosis. Furthermore, mild congestion with a large number of hepatic sinuses narrowed and/or disappeared were observed.

### 3.4. AA-I Impacted Antioxidant System and Aroused Oxidative Stress

SOD levels decreased significantly in all treatment groups in comparison with the CG (*p* < 0.05). Both GSH and T-AOC in the HAG showed significant decreases when compared to the CG (*p* < 0.05), whereas compared with the CG, the results revealed that AA-I-induced MDA and ROS accumulation in the liver is due to the significant elevation of MDA and ROS levels in the HAG (*p* < 0.05). All results are shown in Figure 5.

Furthermore, the relative mRNA levels of antioxidant stress genes were evaluated in the present study. According to the results of QRT-PCR (Figure 6), compared with the CG, the expression of Nrf2, NQO1 and HO-1 mRNA were significantly lower in the MAG and HAG (*p* < 0.05).

### 3.5. AA-I Induced Mitochondrial Damage and Apoptosis in Hepatocytes

The results of mitochondrial damage and hepatocytes apoptosis are shown in Figure 7. The proportions of mitochondrial depolarization were significantly elevated in the HAG as compared with the CG (*p* < 0.05). Compared with the CG, the percentage of hepatocytes apoptosis was significantly elevated in all AA-I treatment groups (*p* < 0.05).

The ultrastructure of hepatocytes is presented in Figure 8. The ultrastructure of hepatocytes in the CG (Figure 8A) appeared normal, with complete mitochondria which have a clear membrane and aligned bridges, intact endoplasmic reticulum, and normal distribution of glycogen particles. The ultrastructure of hepatocytes in the LAG showed swollen mitochondria with decreased electron density in the cytoplasm, and local glycogen particles gathered together (Figure 8B). In the MAG, the electron density in mitochondria was lower and more severe mitochondrial structure pathological changes were observed. Furthermore, hepatocyte apoptosis with concentrated and marginated chromatin was also noticed (Figure 8C,D). It was also noticed that hepatocytes in the MAG showed an indistinct internal structure of cytoplasm, in which large numbers of lysosomes, and glycogen aggregation could be found. In addition, some fuzzy internal structural lumpiness with low electron density, which contained some suspected glycogen and membrane components, were seen. It was observed that the histological changes of hepatocytes in the HAG were similar to those in the MAG. However, the degree of cellular ultrastructure changes was severer than that in the MAG (Figure 8E–H). Furthermore, the hepatocytes in the HAG showed the following histopathological changes: (1) expansion of the mitochondrial outer membrane, (2) cellular vacancy, (3) aggregation of large numbers of secondary lysosomes with uneven electron density, (4) locally, some double-layer membrane circular cavity which contained striated structural materials was present in the liver cytoplasm.

The relative mRNA levels of apoptotic genes involved in the mitochondrial apoptotic pathway were also measured in the present study. As shown in Figure 6, compared with the CG, the mRNA expression of Bcl-2 was apparently down-regulated among all AA-I treatment groups (*p* < 0.05). Additionally, the mRNA expression of Bax increased significantly in all AA-I treatments relative to the control (*p* < 0.05).

## 4. Discussion

Aristolochic acids (AAs) are chemical components which are abundant in traditional Chinese medicines, which have been used commonly as an alternative to antibiotics in the poultry and livestock industries due to the variety of therapeutic effects [27]. However, excessive exposure to AA, especially AA-I, may induce undesired adverse effects [14]. At present, AA-I toxicity has been noticed extensively. It has been proven that the long-term administration of AA-I can be carcinogenic and nephrotoxic in rats, mice, rabbits, pigs and humans [28,29,30]. However, there is scant information regarding the detrimental effects and mechanism of AA-I-induced poultry hepatotoxicity. The present study results reveal that a high dosage of AA-I induces severe hepatotoxicity due to oxidative stress injury, mitochondrial damage and apoptosis of the hepatocytes in Tianfu broilers, which leads to irreversible hepatic dysfunction.

The liver plays an important role of various biological activities, including bile acid secretion, immunization, detoxification and metabolism [31]. It has been reported that histopathological alterations of the liver can be triggered by exposure to microorganisms, metals, drugs, and so on, and result in liver dysfunction [32]. In this study, the results indicated that AA-I can trigger histopathological lesions of hepatocytes in a dose-dependent manner. It has been proven that hepatic parenchyma injury can be recognized by elevation of the liver index [33]. In the present study, elevation of the liver index among all treatment groups can be evidence of liver injury. The levels of serum GOT and GPT are two important indicators of liver function, and the abnormally high levels are generally tested as predictors of liver dysfunction [34,35]. Nevertheless, the significant elevation of serum GOT and GPT levels revealed the true AA-I-induced hepatotoxicity in Tianfu broilers. Overall, the present results indicate that AA-I can cause an increase in the liver index and the elevation of serum GOT and GPT, which is in accordance with the results from Yeh et al.’s study in 2008 [36]. It has been reported that AA-I can induce diffuse hepatocyte swelling and vacuolar degeneration in rats [37]. Similarly, the current result of histopathological changes indicated that AA-I caused hepatocyte injury in a dose-dependent manner and the histopathological changes are characterized by obvious hepatocyte swelling, vesicular degeneration and steatosis.

Although the hepatotoxicity of AA-I has been widely presented in several works, the comprehension of its underlying mechanism remains unknown. It has been indicated that oxidative stress plays an essential role in AA-I-induced liver damage [10,38]. Oxidative stress disrupts the oxidation–reduction balance, which is commonly seen in many biological reactions with increases in ROS levels [39]. The results of this study show that AA-I promoted the accumulation of ROS and MDA, and the suppression of SOD, T-AOC and GSH performance, indicating that continuous exposure to an overdose of AA-I breaks down the redox homeostasis and induces oxidative stress in the hepatocytes of Tianfu broilers. It has been reported that AA-N induces endothelial cell toxicity by the elevation of intracellular ROS levels and inhibition of cellular antioxidant function [40]. Additionally, Zhang et al. [41] indicated that exposure to AA-I perturbed the progression of oocyte meiotic and fertilization capacity by the induction of excessive oxidative stress, which led to DNA damage and apoptosis. It has also been proven that AA-I can induce significant downregulation of the Nrf2-HO-1/NQO1 pathway, an efficient antioxidant, and lead to oxidative stress [42,43].

Oxidative stress is a crucial factor for the induction of apoptosis and excessive ROS promotes cell degradation and death [44]. Romanov et al. [45] reported that a high dose of AA induced renal cell apoptosis by increasing the production of ROS. In the present study, the apoptosis rate of hepatocytes increased significantly in all treatment groups. In order to specify the mechanism of AA-I-induced apoptosis, the biomarkers were strictly limited to the Bcl-2 family (Bcl-2 and Bax) [46]. In the present study, AA-I increased the mRNA expression of Bax, whereas it decreased the expression of Bcl-2. Moreover, it has been reported that the imbalance of the Bax and Bcl-2 ratio and the elevation of mitochondrial depolarization can initiate a mitochondrial apoptotic pathway [47]. Therefore, the significant increase in mitochondrial depolarization in this study implies that AA-I can activate the mitochondria-regulated apoptotic signal transduction. Furthermore, the findings of hepatocyte ultrastructure changes also prove the occurrence of cellular apoptosis. It has been revealed in several studies that AA-I-induced ROS accumulation can cause the disruption of mitochondria. The mitochondria became swollen with decreased electron density and the mitochondrial transmembrane potentially collapses, leading to mitochondrial dysfunction [48]. Furthermore, numerous electrons leaked out from the uncoupled electron transport chain following the process of uncoupled oxidative phosphorylation in energy respiration, thereby resulting in large amounts of ROS generation, and further arousing explosive oxidative stress [49,50]. Liu et al. [51] have proven that mitochondrial dysfunction is involved in AA-I-induced apoptosis in renal proximal tubular epithelial cells. Nevertheless, the disorder of mitochondrial microstructure and function are associated with AA-I-induced hepatocyte apoptosis and oxidative damage.

## 5. Conclusions

To conclude (as summarized in Figure 9), the current work demonstrates that AA-I induced hepatotoxicity after subchronic exposure in Tianfu broilers in a dose-dependent manner. It breaks the redox balance of the liver, which further triggers oxidative-stress-mediated apoptosis and mitochondrial damage, eventually leading to liver damage.

## Figures and Tables

**Figure 1 animals-11-03437-f001:**
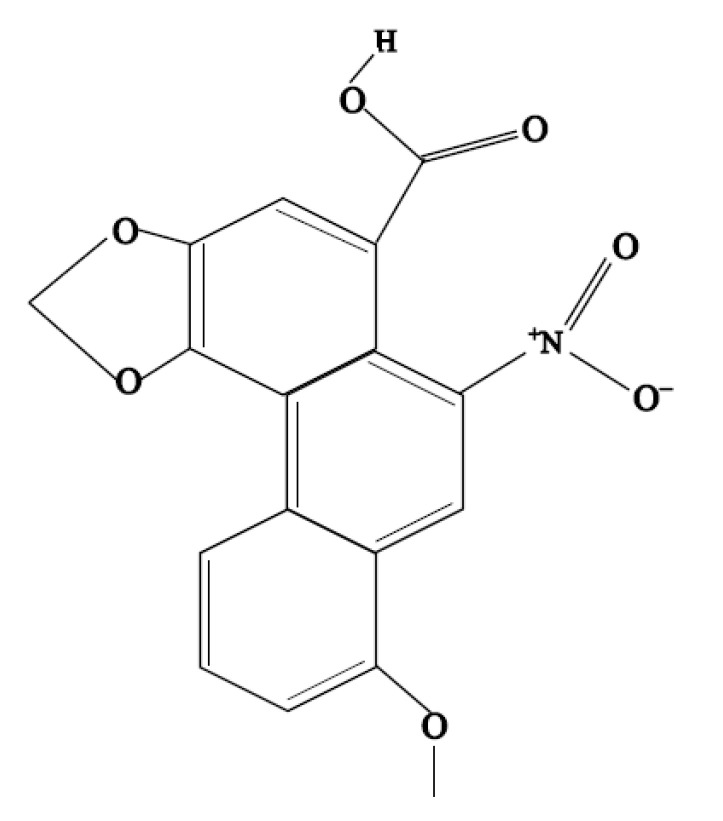
Chemical structure of aristolochic acid A.

**Figure 2 animals-11-03437-f002:**
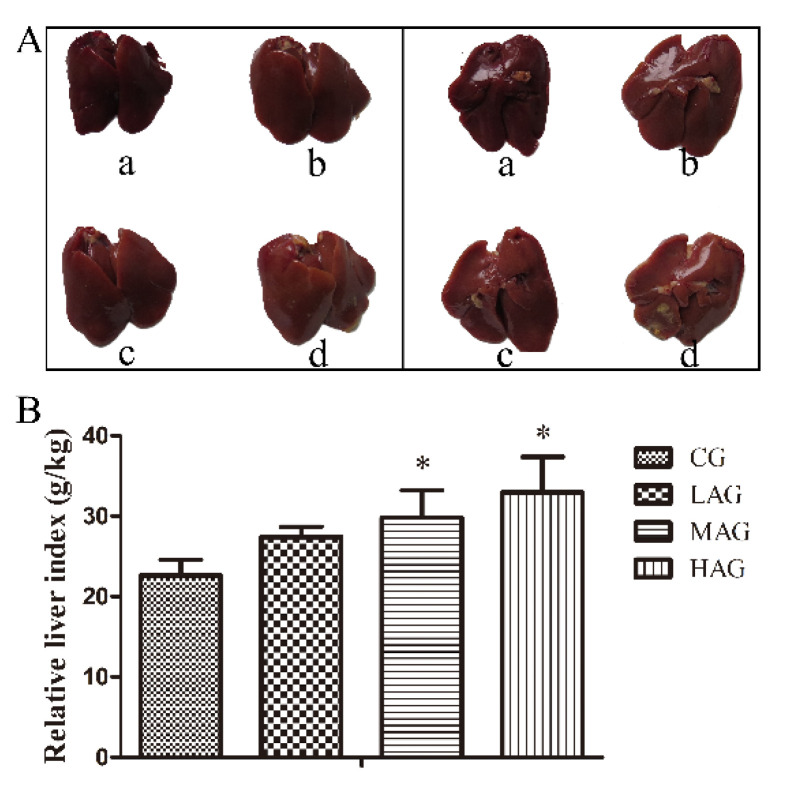
(**A**): Liver morphology in different groups. a: CG group, b: LAG group, c: MAG group, d: HAG group. (**B**): Liver index of broilers in different groups. * *p* < 0.05, compared with the control group (CG).

**Figure 3 animals-11-03437-f003:**
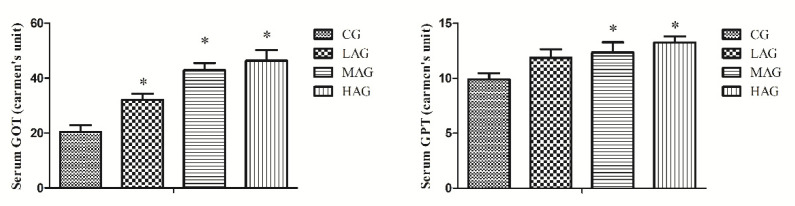
Effects of aristolochic acid A on the liver function of Tianfu broilers. * *p* < 0.05, compared with the control group (CG). GOT: glutamic oxalacetic transaminase (μmol/L), GPT: glutamic-pyruvic transaminase (mmol/L).

**Figure 4 animals-11-03437-f004:**
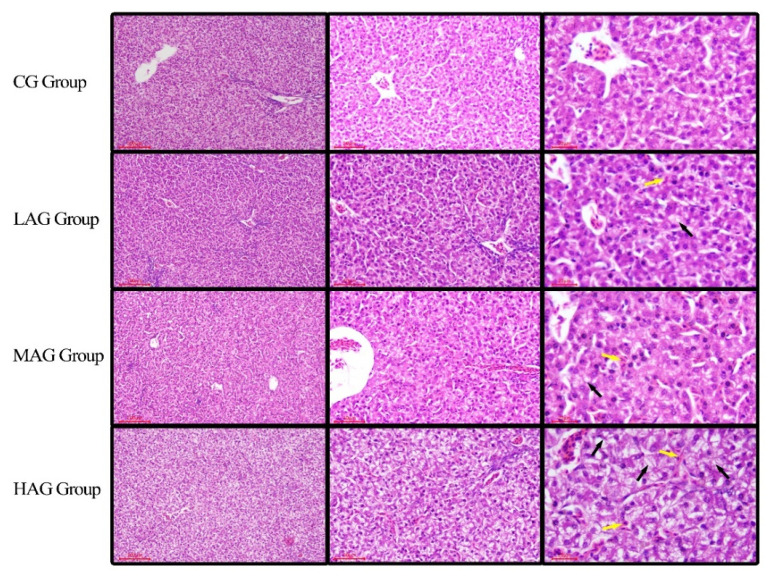
The histopathological changes of hepatocytes induced by AA-I in Tianfu broilers. H.E. stain, the left magnification: 100 μm, the medium magnification: 50 μm, the right magnification: 20 μm. Yellow arrows: red blood cells; black arrows: hepatocyte vesicular degeneration.

**Figure 5 animals-11-03437-f005:**
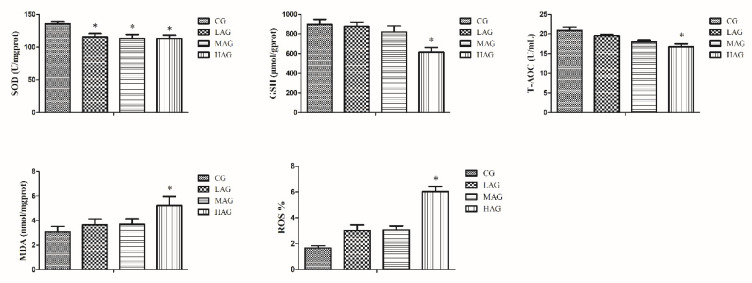
Effects of aristolochic acid A on the antioxidant system of Tianfu broiler. * *p* < 0.05, compared with the control group (CG). MDA: malondialdehyde (mmol/L), SOD: superoxidase dismutase (U/mL), GSH: glutathione (μmol/g), and T-AOC: total antioxidant capacity (U/mL); ROS: reactive oxygen species (%).

**Figure 6 animals-11-03437-f006:**
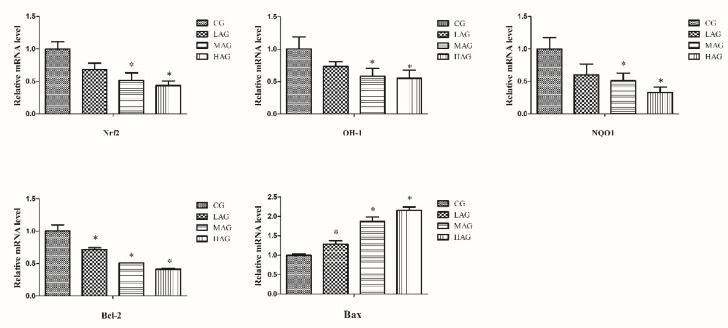
The mRNA expression of genes (NQO1, HO-1, Nrf2, Bax, and Bcl-2) of liver in each group. * *p* < 0.05, compared with the control group (CG).

**Figure 7 animals-11-03437-f007:**
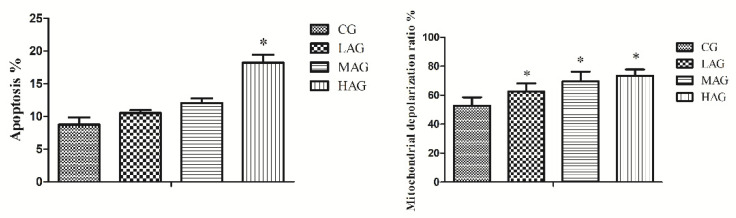
The mRNA expression of genes (NQO1, HO-1, Nrf2, Bax, and Bcl-2) of livers in each group. * *p* < 0.05, compared with the control group (CG).

**Figure 8 animals-11-03437-f008:**
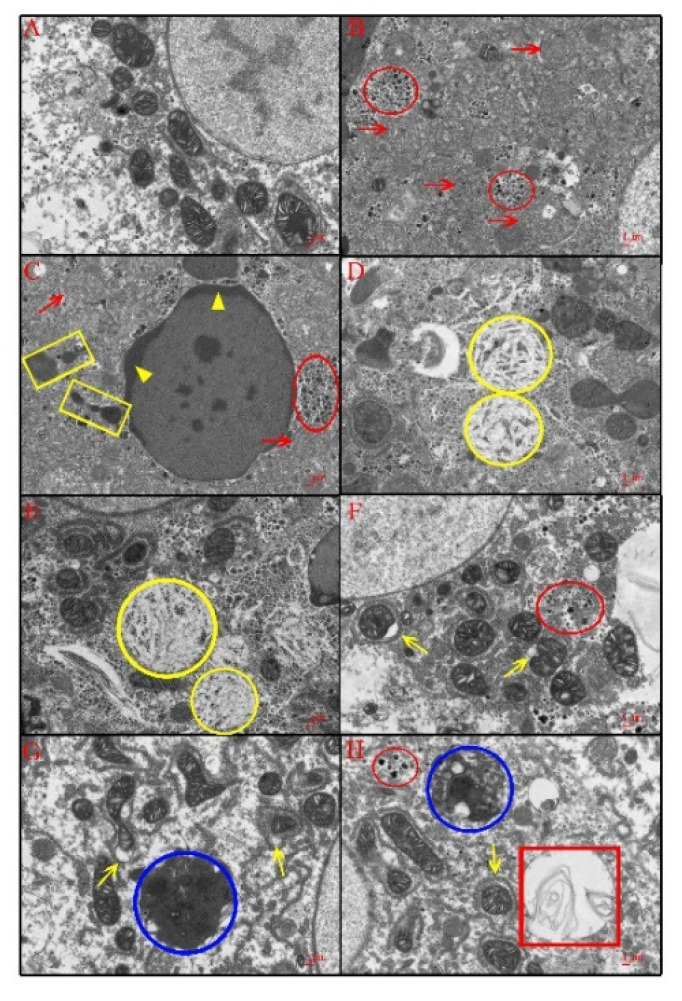
Ultrastructure observations of livers in different groups. (**A**): CG group; (**B**): LAG group; (**C**,**D**): MAG group; (**E**–**H**): HAG group; scale bar = 10 μm. Red circle: glycogen aggregation; yellow circle: fuzzy internal structural lumpiness with low electron density; blue circle: secondary lysosomes with uneven electron density; yellow triangle: the chromatin concentrated and shifted in the nucleus; red arrow: swollen mitochondria; yellow arrow: the mitochondria outer membrane expanded, locally forming a vacancy; red box: double-layer membrane circular cavity containing striated structure material; yellow box: the number of lysosomes increased significantly.

**Figure 9 animals-11-03437-f009:**
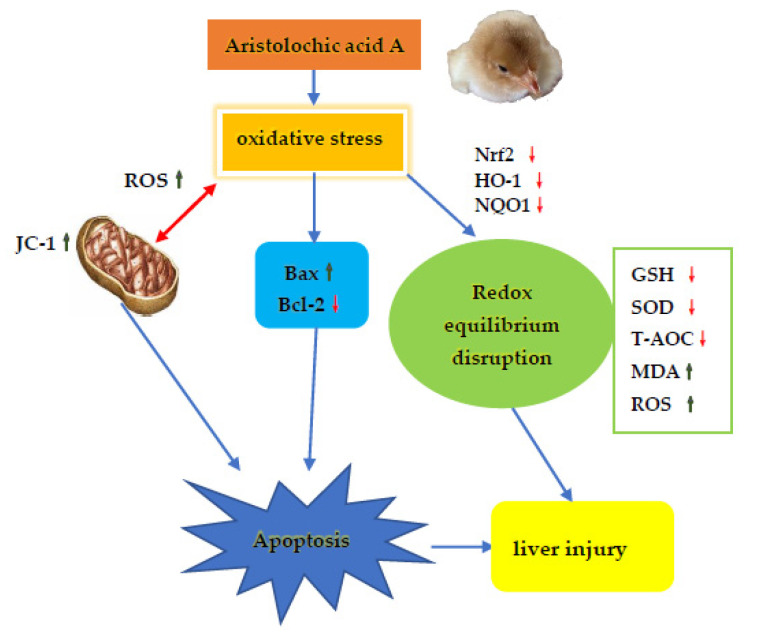
Schematic diagram of the possible mechanism of AA-I-induced liver injury in Tianfu broilers. The toxicological mechanism of AA-I-induced liver injury contains excessive apoptosis and oxidative stress damage.

**Table 1 animals-11-03437-t001:** Illustration of a 4-point scoring system.

Numerical Score	Description	Definition
0	Within normal limits	Tissue considered to be normal, under the conditions of the study and considering the age, sex, and strain of the animal concerned. Alterations may be present, which, under other circumstances, would be considered deviations from normal.
1	Minimal	The amount of change present barely exceeds that which is considered to be within normal limits.
2	Slight	In general, the lesion is easily identified but of limited severity.
3	Moderete	The lesion is prominent, but there is significant potential for increased severity.
4	Severe	The degree of change is as complete as possible (occupies the majority of the organ).

## Data Availability

The data presented in this study are available on request from the corresponding authors. The data are not publicly available due to privacy protection.

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
