# Peer review of "Aristolochic Acid I-Induced Hepatotoxicity in Tianfu Broilers Is Associated with Oxidative-Stress-Mediated Apoptosis and Mitochondrial Damage"

_animals, 2021, doi:10.3390/ani11123437_

Round 1

Reviewer 1 Report

In the present paper I carefully reviewed, the Authors have described the mechanism of action of Aristolochic Acid I (AA-I) induced hepatotoxicity, including oxidative stress, oxidative stress-mediated apoptosis and mitochondrial damage, in order to provide a guidance of application of AA-I in domestic poultry husbandry.

I would like to congratulate Authors for the good-quality of their article, the literature reported used to write the paper, and for the clear and appropriate structure.

The manuscript is well written, presented and discussed, and understandable to a specialist readership.

In general, the organization and the structure of the article are satisfactory and in agreement with the journal instructions for authors. The subject is adequate with the overall journal scope.

The work shows a conscientious study in which a very exhaustive discussion of the literature available has been carried out.

The Introduction section provides sufficient background (however, additional references of recently published papers may add value to this section), and the other sections include results clearly presented and analyzed exhaustively.

However, as specific comments, with the aim to further improve the quality of the paper, the Conclusion section could be improved; also, the Authors have to check if alle references have been cited in the text.

A couple of additional recently published references may add value to the Introduction section.

Also, add the appropiate references for:

2.3.2. Biochemical analysis

2.3.4. Reactive Oxygen Species (ROS), Mitochondrial Membrane Potential (MMP) and Apoptosis by Flow Cytometry (add also the kits used)

Finally, check the references style.

Author Response

Response to Reviewer 1 Comments

In the present paper I carefully reviewed, the Authors have described the mechanism of action of Aristolochic Acid I (AA-I) induced hepatotoxicity, including oxidative stress, oxidative stress-mediated apoptosis and mitochondrial damage, in order to provide a guidance of application of AA-I in domestic poultry husbandry.

I would like to congratulate Authors for the good-quality of their article, the literature reported used to write the paper, and for the clear and appropriate structure.

The manuscript is well written, presented and discussed, and understandable to a specialist readership.

In general, the organization and the structure of the article are satisfactory and in agreement with the journal instructions for authors. The subject is adequate with the overall journal scope.

The work shows a conscientious study in which a very exhaustive discussion of the literature available has been carried out.

The Introduction section provides sufficient background (however, additional references of recently published papers may add value to this section), and the other sections include results clearly presented and analyzed exhaustively.

We are grateful for your carefully comments and suggestions. We have revised the manuscript animals-1457248 in according with the comments and suggestions one by one. The changes we have made to the manuscript are as following and shown in modified vision with changing track. Hopefully, our manuscript is now acceptable for publication in Animals.

Point 1: However, as specific comments, with the aim to further improve the quality of the paper, the Conclusion section could be improved; also, the Authors have to check if alle references have been cited in the text.

Response 1: Thanks for your great comment, and we had revised as suggested. We re-built the conclusion part, and also, we had checked and modified all the references that had been cited in the text.

Point 2: A couple of additional recently published references may add value to the Introduction section.

Response 2: Thanks for your great comment, and we had revised as suggested. We had cited some recently published references in the introduction part.

Point 3: Also, add the appropiate references for:

2.3.2. Biochemical analysis

2.3.4. Reactive Oxygen Species (ROS), Mitochondrial Membrane Potential (MMP) and Apoptosis by Flow Cytometry (add also the kits used)

Response 3: Thanks for your great comment, and we had revised as suggested. And we added the kits used for Reactive Oxygen Species (ROS), Mitochondrial Membrane Potential (MMP) and Apoptosis in the Experimental materials part.

Point 4: Finally, check the references style.

Response 4: Thanks for your great comment, and we had checked and modified the references style thru through the whole manuscript.

Reviewer 2 Report

In general the reviewed manuscript is very interesting and generally well designed. Some aspects need to be clarified before the publication.

  1. How many researchers were involved in the assessment of macro, micro and ultrascopic liver/hepatocytes morphology ? Did they were blind to the groups ?
  2. Line 200 – the authors wrote “moderate histopathological lesion”. In line 248 – “degree of cellular ultrastructure changes were severer”. To judge these changes semi-quantitative scales are needed. Did the authors use any? If yes, what were the criteria?
  3. In Figure 2A there is an information about the “renal morphology”
  4. Figure 4 – the description is misleading. The section is not pathological itself, the liver is pathological.
  5. Conclusion is poorly written. It simply reiterates the obtained results. The authors declared that the aim of the study was “to explore the mechanism of action of AA-I induced hepatotoxicity”. To my surprise, I see no any explanation of this mechanism I only see morphological consequences of the liver intoxication.

Author Response

Response to Reviewer 2 Comments

In general, the reviewed manuscript is very interesting and generally well designed. Some aspects need to be clarified before the publication.

We are grateful for your carefully comments and suggestions. We have revised the manuscript animals-1457248 in according with the comments and suggestions one by one. The changes we have made to the manuscript are as following and shown in modified vision with changing track. Hopefully, our manuscript is now acceptable for publication in Animals.

Point 1: How many researchers were involved in the assessment of macro, micro and ultrascopic liver/hepatocytes morphology? Did they were blind to the groups?

Response 1: Thanks for your great comment. Actually, two researchers were involved in the assessment of macro, micro and ultrascopic liver/hepatocytes morphology, and they had been listed as authors for this work. And yes, the researchers were blind to the groups. We had re-built the Author Contributions part, and added the information of who did this contribution. Moreover, in the ACKNOWLEDGE section, we added related information to thank another two pathologists who evaluated the slices.

Point 2: Line 200 – the authors wrote “moderate histopathological lesion”. In line 248 – “degree of cellular ultrastructure changes were severer”. To judge these changes semi-quantitative scales are needed. Did the authors use any? If yes, what were the criteria?

Response 2: Thanks for your great comment. In pathological diagnosis, we used INHAND criteria to judge the severity of histological lesions, and we had cited it in the Material and Methods part. So, the moderate histopathological lesion means that “the lesion is prominent, but there is significant potential for increased severity”. In this sentence, “degree of cellular ultrastructure changes were severer” descripted the changes compared with that of the MAG.

Point 3: In Figure 2A there is an information about the “renal morphology”

Response 3: Thanks for your great comment, and we had revised as suggested.

Point 4: Figure 4 – the description is misleading. The section is not pathological itself, the liver is pathological.

Response 4: Thanks for your great comment, and we had revised as suggested.

Point 5: Conclusion is poorly written. It simply reiterates the obtained results. The authors declared that the aim of the study was “to explore the mechanism of action of AA-I induced hepatotoxicity”. To my surprise, I see no any explanation of this mechanism I only see morphological consequences of the liver intoxication.

Response 5: Thanks for your great comment, and we had re-built the conclusion part as suggested. Based on the results of this study, AA-I induced hepatotoxicity in Tianfu broilers by inducing oxidative stress damage with excessive ROS generation along with redox equilibrium disruption which was evidenced by the decreased expressions of NQO1, HO-1, and Nrf2. And this process further induces excessive apoptosis by elevating the expression of Bax and decreasing the expression of Bcl-2. Moreover, the mitochondrial damage aroused by oxidative stress also contributes to excessive apoptosis.

Round 2

Reviewer 1 Report

The revised version of the paper is suitable for the final acceptance.

Reviewer 2 Report

I accept the explanations and corrections presented by the authors.